# MRI and CT Fusion in Stereotactic Electroencephalography: A Literature Review

**Jaime Perez [1,\*], Claudia Mazo [1,2,3], Maria Trujillo [1] and Alejandro Herrera [1,4]**

1.  Multimedia and Computer Vision Group, Universidad del Valle, Cali 760001, Colombia; claudia.mazovargas@ucd.ie (C.M.); maria.trujillo@correounivalle.edu.co (M.T.); herrera.alejandro@correounivalle.edu.co (A.H.)
2.  UCD School of Computer Science, University College Dublin, Dublin 4, Ireland
3.  CeADAR Ireland's Centre for Applied AI, Dublin 4, Ireland
4.  Clinica Imbanaco Grupo Quironsalud, Cali 760001, Colombia
\*  Correspondence: perez.jaime@correounivalle.edu.co

**Abstract:** Epilepsy is a common neurological disease characterized by spontaneous recurrent seizures. Resection of the epileptogenic tissue may be needed in approximately 25% of all cases due to ineffective treatment with anti-epileptic drugs. The surgical intervention depends on the correct detection of epileptogenic zones. The detection relies on invasive diagnostic techniques such as Stereotactic Electroencephalography (SEEG), which uses multi-modal fusion to aid localizing electrodes, using pre-surgical magnetic resonance and intra-surgical computer tomography as the input images. Moreover, it is essential to know how to measure the performance of fusion methods in the presence of external objects, such as electrodes. In this paper, a literature review is presented, applying the methodology proposed by Kitchenham to determine the main techniques of multi-modal brain image fusion, the most relevant performance metrics, and the main fusion tools. The search was conducted using the databases and search engines of Scopus, IEEE, PubMed, Springer, and Google Scholar, resulting in 15 primary source articles. The literature review found that rigid registration was the most used technique when electrode localization in SEEG is required, which was the proposed method in nine of the found articles. However, there is a lack of standard validation metrics, which makes the performance measurement difficult when external objects are presented, caused primarily by the absence of a gold-standard dataset for comparison.

**Keywords:** image fusion; stereotactic electroencephalography; computer tomography; magnetic resonance imaging; image registration

## 1. Introduction

Epilepsy is a neurological disease affecting approximately 50 million people worldwide [1]. Between 25% and 30% of cases are untreatable with anti-epileptic drugs [2,3]. In those cases, resection of the seizure focus area may be necessary [4].

The resection surgery for pharmacoresistant epilepsy relies on the correct detection of the epileptogenic tissue [5,6]. The detection depends on invasive diagnostic techniques such as Stereotactic Electroencephalography (SEEG) [7]. SEEG measures electric signals using deep electrodes implanted in the brain. The implantation is guided using a Magnetic Resonance Image (MRI) with a stereotactic frame affixed to the head prior to the implantation. After the implantation, a Computer Tomography (CT) image is taken to obtain the localization of the electrodes, and finally, an image fusion is performed between the pre-implantation MRI and the post-implantation CT. Image fusion is a powerful technique because it synthesizes the localization of the electrodes and the structural anatomical information in a single image [8,9]. However, the presence of external objects may affect the performance of fusion techniques. The literature review focuses primarily on fusion techniques between images of MRI and CT, with special attention on the software tools, evaluation metrics, and presence of external objects.

The image fusion (Figure 1) maps images into the same coordinate system and then blends the aligned result into an output image. Among the methods for image fusion, characteristics such as imaging modalities determine the performance of a procedure. For example, methods that use the Mean-Squared Difference (MSD) as the optimization metric perform better in single-modality fusion [10,11]. Thus, it is essential to know the performance of these techniques, especially in applications such as SEEG that involve external objects.

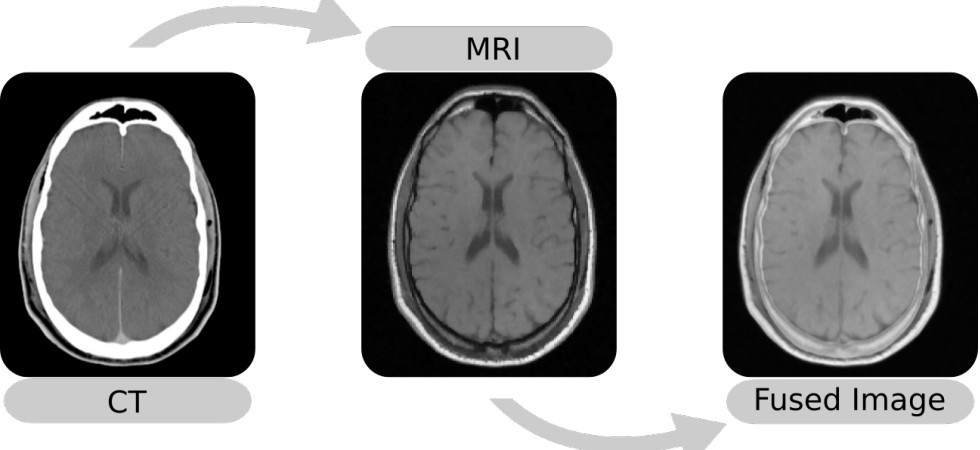

**Figure 1.** Example of multi-modal image fusion between an MRI and a CT. The images were taken from Patient 001 of the RIRE dataset [12].

This paper aims to present an overview of the main methods, tools, metrics, and databases used in multi-modal image fusion. This review contributes to the literature by summarizing the main techniques of image fusion, distinguishing between the two main steps of registration and merging.

## 2. Materials and Methods

The literature review followed the methodology proposed by Kitchenham [13], which is specific to software engineering. Moreover, Kitchenham's method aligns with the PRISMA methodology [14]. Table 1 illustrates the section defined by Kitchenham for systematic literature reviews and the equivalent to the PRISMA checklist.

**Table 1.** The Kitchenham methodology vs. the PRISMA methodology.

| Kitchenham Section | PRISMA Section |
| --- | --- |
| Title | Title |
| Executive summary or structured abstract | Abstract |
| Background and review questions | Introduction |
| Data sources | Information sources |
| Search strategy and included and excluded studies | Search strategy |
| Study selection and quality assessment | Selection process |
| Data extraction | Data collection process and data items |
| Data synthesis | Synthesis methods |
| Results | Results |

The review included three steps: First the review planning was conducted using the research questions. Second, the review was conducted by applying the search strategy. Third, the collected information was synthesized to answer the research questions. The next section presents details on the review stages.

### 2.1. Planning

Based on the necessity for multi-modal image fusion in diagnostic exams with external objects, such as SEEG, where errors in the fusion could lead to inaccurate detection of the epileptogenic tissue, the following research questions were proposed, focusing mainly on the multi-modal fusion between CT and MRI:

(i)     What are the existing methods of brain image fusion using CT and MRI?
(ii)    What are the tools used to fuse multi-modal brain images?
(iii)   What are the metrics used to validate and compare image fusion methods?

### 2.2. Conducting the Review

The databases and search engines of Scopus, IEEE, PubMed, Springer, and Google Scholar were useful to conduct the review, because they include articles from the medical and engineering area. The following keywords were used to select articles about multi-modal brain image fusion: "image registration", "image fusion", "medical imaging", "brain", "neuroimaging", "computer tomography", "CT", "magnetic resonance imaging", and "MRI". The dates of the selected articles were between January 2010 and April 2021, obtaining a total of 1111 papers in the first stage. Afterwards, the following inclusion criteria were used: (i) studies about multi-modal image fusion or registration; (ii) studies that used brain images; and (iii) studies that specified the registration and merging method. This stage resulted in a total of 361 papers. Finally, two exclusion criteria were used: (i) studies without validation of the procedure; and (ii) studies that did not use CT or MRI. The final stage returned a total of 15 papers (Table 2 and Figure 2).

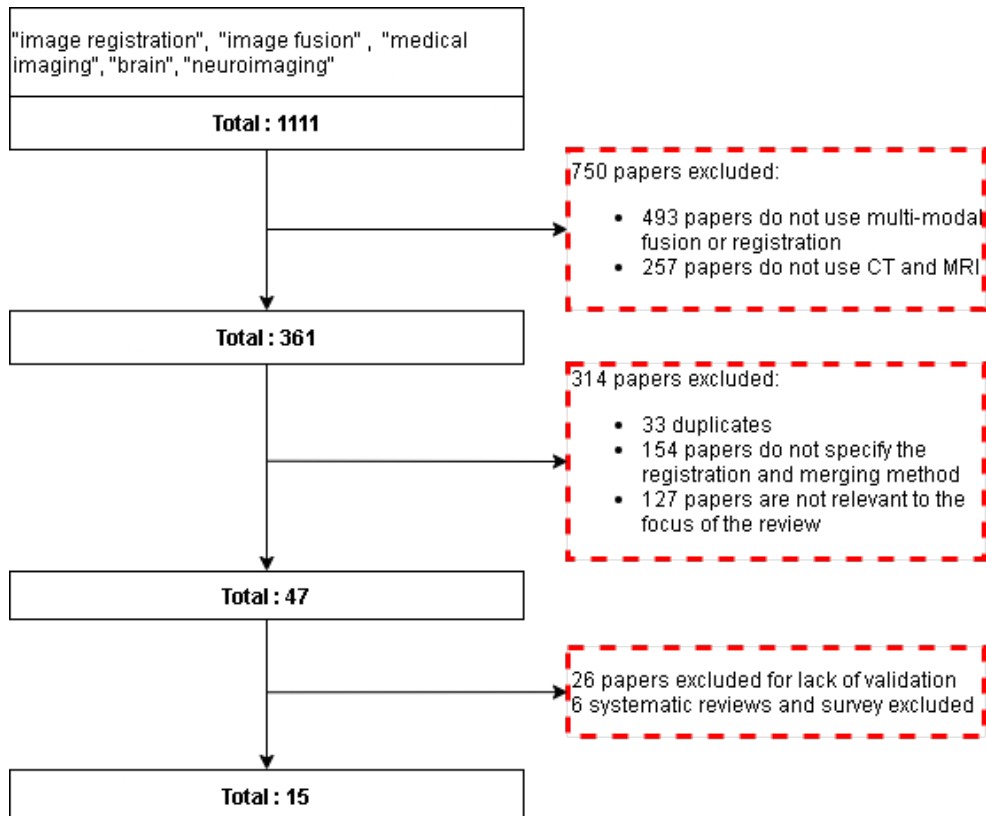

**Figure 2.** Flow diagram for the literature review. Red dotted squares represent the excluded papers.

**Table 2.** The fifteen original research articles selected.

| Paper | Method | Image Source | Validation | Software Platform |
|-------|--------|--------------|------------|-------------------|
| [15] | Rigid image registration | Retrospective Image Registration Evaluation (RIRE) | Target Registration Error (TRE) | Elastix ITK |
| [16] | Rigid Image registration | 2D slices from the RIRE | Rotation and translation error | MATLAB |
| [17] | Rigid image registration | Image from ten patients with pharmacoresistant epilepsy | Error in the localization of electrodes measured in mm | ITK |
| [18] | Non-rigid image registration | Image from five patients with pharmacoresistant epilepsy | Error in the localization of electrodes measured in mm | BrainLab |
| [19] | Rigid image registration | Image from 20 datasets of Deep Brain Stimulation (DBS) studies | Target area registration error | C++ |
| [20] | Rigid image registration | CT and MRI images from seven patients with epilepsy. | Error in the localization of electrodes measured in mm | MATLAB SPM |
| [21] | Rigid image registration | Image from five patients with pharmacoresistant epilepsy | Euclidean distance between the location estimated procedure and determined by visual inspection | MATLAB SPM |
| [22] | Automatic method to evaluate and quantify the multi-modal image registration accuracy | Porcine skull dataset and RIRE | Electrode localization accuracy measured in mm | MATLAB |
| [23] | Quantify the registration accuracy in SEEG, from pre-implantation MRI and post-implantation CT | Image from 14 patients with pharmacoresistant epilepsy | Error in the localization of electrodes measured in mm | FSL |
| [24] | Multi-modal image fusion | Two image datasets of CT and MRI | Peak-Signal-to-Noise-Ratio (PSNR), Mean-Squared Error (MSE), and Entropy (EN) | N/A |
| [25] | Multi-modal image fusion | Nine pairs of MRI and CT images, from patients with severe cardiovascular accident | Mutual Information (MI), Standard Deviation (STD), Universal Image Quality Index (UIQI), and Spatial Frequency (SF) | MATLAB |
| [26] | Multi-modal image fusion using Pulse-Coupled Neural Network (PCNN) | five pairs of MRI and CT images | MI, SF, STD, EN, and Structural Similarity Index (SSIM) | MATLAB |
| [27] | Multi-modal image fusion using WT and neuro-fuzzy | Two pairs of MRI and CT images | EN, MI, and fusion factor | N/A |
| [28] | Multi-modal image fusion using Discrete Wavelet Transform (DWT) and Guider Filter (GF) | 14 pairs of MRI and CT images | STD, average gradient, and edge strength | N/A |
| [29] | Multi-modal image fusion using Non-Subsampled Shearlet Transform (NSST) | 10 pairs of medical grayscale images and 4 pairs of medical color images | EN, STD, MI, SSIM, and edge strength | MATLAB |

Three criteria were used to evaluate the quality of the papers: (i) complete description of the image fusion or registration method; (ii) complete description of the validation; and (iii) complete description of the databases used for validation.

## 3. Results

This section is divided into three parts according to the research questions (Section 2.1): methods of image fusion (Section 3.1), performance metrics (Section 3.2), and image fusion tools (Section 3.3).

### 3.1. Methods of Image Fusion

Image fusion is a technique that composes an image with better information from multiple inputs, which has two steps: registration and merging. The first step transforms all input images into a common standard to represent the same object or phenomena, while the second step merges all aligned images [30]. The article was organized taking the image registration and image merging separately.

### 3.1.1. Image Registration

Image registration consists of finding a transformation $T$ such that it aligns a moving image $I_M$ with a fixed image $I_F$, by minimizing a cost function $C$, where the vector $\mu$ contains the parameters for the transform (Equation (1)) [31,32].

$$\mu = arg \min_{\mu} C(T_\mu; I_F, I_M). \tag{1}$$

Figure 3 summarizes the procedure where a transformation was applied to $I_M$. Then, a similarity metric between $I_M$ and $I_F$ was optimized by updating the transform.

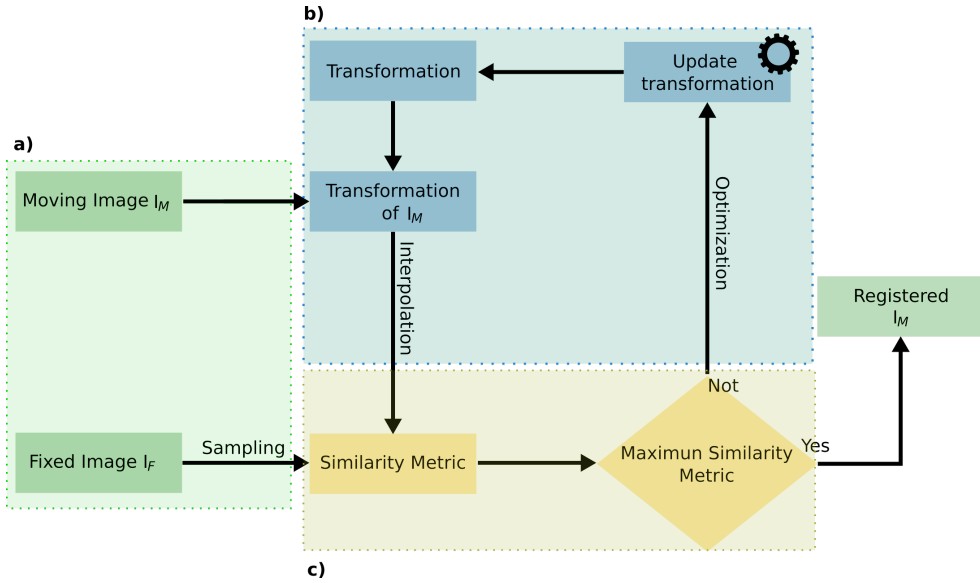

**Figure 3.** Registration procedure. (**a**) Input images for the registration. (**b**) Optimization of the transformation for $I_M$. (**c**) Measurement of the similarity metric to compare the transformed $I_M$ with $I_F$.

The registration procedure could be classified based on two criteria: (i) the transformation (Figure 3b), where the procedure was sub-classified into rigid and non-rigid; (ii) the optimization metric, where the registration was sub-classified according to the similarity metric (Figure 3c). Considering these criteria, first, the concept of transformation and similarity metrics is presented, and then, the results are summarized.

Registration: Transformation

The first type of method is the rigid registration, which applies a single transformation to the entire $I_M$. This method can be categorized, depending on the transformation type and degrees of freedom, into [31]:

(i)    translation: translation only (three degrees of freedom in 3D);
(ii)   Euler: rotation and translation (six degrees of freedom in 3D);
(iii)  similarity: rotation, scaling, and translation (nine degrees of freedom in 3D);
(iv)   affine: translation, rotation, scaling, and shearing (12 degrees of freedom in 3D).

These transformations define the degrees of freedom of the deformation model. The affine transformation has the most degrees of freedom in comparison.

The second type of method is the non-rigid registration, which are computationally more complex than rigid registration, since this requires finding local deformations of the moving image. Non-rigid methods are sub-classified into two main categories based on the deformation model: transformations derived from physical models and transformation derived from interpolation and approximation models [33].

Registration: Metrics

The metric or cost function is an essential part of the registration procedure, which also determines the type of transformation and optimization [32,34]. The most common metrics are:

(i)　Mean-Squared Difference (MSD): measures the average gray difference between the $I_F$ and $I_M$ voxels;
(ii)　Correlation Coefficient (CC): measures the gray-level similarities between the $I_F$ and $I_M$ voxels [35];
(iii)　Normalized Correlation Coefficient (NCC): is the CC after normalizing the images;
(iv)　Mutual Information (MI): measures the dependency between two variables (moving and fixed image) and is calculated using the entropy, the probability joint histogram distribution, and the Parzen window estimation [35];
(v)　Normalized Mutual Information (NMI): an alternative to the MI metric that solves the drawbacks such as misregistration in small overlapping regions [10].

The most common metrics in multi-modal brain registration are the MI and the NMI, which rely on the probability distribution between the images [10,11].

Registration Methods Literature

Among the 15 selected original research articles, seven proposed methods for rigid registration (Table 3). Taimouri's method is an example of a non-rigid registration method with the use of the MI to localize deep electrodes in SEEG [17]. This method uses the Insight ToolKit (ITK), an open-source toolkit for medical image registration and segmentation [36]. Finally, the method was validated using images of ten patients, five males and five females, aged between eight and seventeen years. For the accuracy measurement, a photograph of the electrodes was taken from a digital camera, and then, the electrodes of the resulting MRI were projected in the 2D photograph. The accuracy was the error between the electrodes from the photography and the electrodes from the 3D MRI. While the method yielded a low average localization error of 1.31 ± 0.69 mm for 385 electrodes, the accuracy measurement procedure did not consider the error of the projection, which implies uncertainty if the error comes from the registration or the projection procedure.

**Table 3.** Articles about multi-modal image registration methods.

| Paper | Transform | Optimization Metric | Optimization Method |
|---|---|---|---|
| [15] | Rigid | Mutual information, with curvelet-based sampling | Gradient descent |
| [16] | Rigid | Interaction energy and the mutual information | Genetic Algorithm (GA) |
| [17] | Rigid | Mutual information | Powell's method |
| [18] | Non-rigid | N/A | N/A |
| [19] | Rigid | Normalized Gradient Fields (NGFs) | Multilevel Gauss–Newton approach |
| [20] | Rigid | Normalized Mutual Information (NMI) | Gradient descent |
| [21] | Rigid | Mutual information | Gradient descent |

Stieglitz et al. [18] used non-rigid transformation with MRI and CT to evaluate if the registration improved the localization of the SEEG electrodes. For the registration, Stieglitz used BrainLab iPlan, which is a radiosurgery planning software for rigid registration, and for non-rigid registration, the Automated Elastic image Fusion algorithm (AEF) and the Guided Elastic image Fusion algorithm (GEF) were used. The non-rigid registration improved the fusion, but there were unclear results in the localization of the electrodes, caused mainly by the evaluation method, which measured the localization performance using the distance between the electrodes and the brain cortex and not the real position of

the electrodes in the brain. Another cause for the unclear results was the lack of images for the validation, which was present in other studies that used intracranial electrodes, such as the method proposed by Dykstra et al. [21] or the method proposed by Hermesa et al. [20], which used the MI and the NMI, respectively, for the registration.

The MI and the NMI also have some drawbacks due to Shannon's entropy, which assumes voxel value independence, but images lack this characteristic. However, a change in the sampling reduces this problem by selecting the less dependent voxel. With this premise, Freiman et al. [15] in 2011 proposed a sampling method using the curvelet transform, which is an extension of the wavelet transform, more suitable for two-dimensional and three-dimensional signals. Freiman et al. used a sampling method with a rigid registration, using the Retrospective Image Registration Evaluation database (RIRE), a public database used to validate registration methods. The method yields better results in the Target Registration Error (TRE) compared to methods based on uniform and gradient-based sampling.

Some authors have used metrics different from the previous ones, such as the multi-modal registration procedure proposed by Panda et al. [16]. Panda et al.'s method uses Evolutionary Rigid-Body Docking (ERBD) algorithms, which is a docking technique that predicts the optimal configuration between two molecules. This registration takes the input images as molecules and minimizes the energy between the images using evolutionary algorithms. The ERBD composes the energy from two metrics: the interaction energy and the MI. Panda et al. developed this method using MATLAB, and it only works with 2D images, causing difficulties in the validation for the use of only a few slices of a patient from the RIRE dataset. For clinical use, this method requires a further development and validation for 3D images.

Another example of a registration procedure using a metric different from the MI is the method proposed by Rühaak et al. [19]. This method uses the Normalized Gradient Field distance (NGF), which works on the assumption that two images are similar if a change in intensity happens in the same locations (edges). Rühaak et al. developed this method using C++ and validated it with images from 20 DBS studies, measuring the error against the manual registration with three experts, yielding an average registration error of $0.95 \pm 0.29$ mm. The proposed method by Rühaak et al. can be implemented in other studies that use external objects such as SEEG, and the validation method can be applied to evaluate the registration performance. However, this validation method could lead to subjective error due to the manual registration.

The disadvantages of the MI are the susceptibility to local minimum convergence and the joint density calculation complexity. Computing the smoothest cost function with different kernels in the Parzen window probability estimation reduces the first problem [10]. The last problem can be decreased using a different similarity metric, such as the Normalized Gradient Fields (NGFs), which measure the angle between two image gradients in specific locations. The NGF is faster to compute and has a similar performance to the MI registration methods [37,38].

### 3.1.2. Image Merging Methods

Merging is the process of combining the aligned images. These methods are classified into three categories: Multi-Scale (MSD), non-Multi-Scale (non-MSD), and hybrids [39]. This section presents an overview of these methods. Table 4 shows the found articles about emerging methods.

**Table 4.** Articles about multi-modal image merging.

| Paper | Merging Category | Method |
| --- | --- | --- |
| [24] | MSD | Image fusion using Discrete Wavelet Transform (DWT) and Discrete Ripple Transform (DRT); |
| [25] | MSD | Image fusion using Non-Subsampled Shearlet Transform (NSST) |
| [26] | Non-MSD | Pulse-Coupled Neural Network (PCNN) |
| [27] | Hybrid | Multi-modal image fusion using WT and neuro-fuzzy |
| [28] | MSD | Fusion using Discrete Wavelet Transform (DWT) and Guider Filter (GF) |
| [29] | MSD | Image fusion using Non-Subsampled Shearlet Transform (NSST) |

Multi-Scale Decomposition Methods

These are methods that transform an input image $I_k, k \in \{1, 2, ..., K\}$ into a multi-scale representation $y_l^k, l \in \{1, 2, ..., L\}$ [30]. The most common techniques are the pyramidal and transform domain methods [39]. The first method decomposes the image into an array of different scales, then combines decomposed images using fusion rules. The Laplacian Pyramid Transform (LPT) and the local Laplacian Filter (LLF) are examples of pyramidal methods. In contrast, the domain transform methods decompose an image into lower approximations, then fuses every approximation into a single image. Examples of these techniques are: (i) the Discrete Wavelet Transform (DWT); (ii) the Discrete Ripple Transform (DRT); and (iii) the Non-Subsampled Shearlet Transform (NSST).

An example of a Multi-Scale Decomposition (MSD) is the proposed technique by Patel et al., which combines the DWT and the DRT, which is robust to discontinuities in edges and contours [24]. This method consists of the following steps:

1. Taking input images, specifically CT and MRI, and aligning them to the same magnitude;
2. Applying the DWT to align images;
3. Obtaining the wavelet coefficient map from the aligned images;
4. Applying the DRT to the wavelet coefficient maps and obtaining an initial fused image;
5. Applying the inverse DWT and obtaining the fused image.

Patel validated the algorithm against DWT methods using two pairs of MRI and CT images. This validation uses the metric of the Root-Mean-Squared Error (RMSE) and Peak-Signal-to-Noise-Ratio (PSNR), where Patel's method exhibited the best performance with a PSNR of 20.56 and an RMSE of 572.23. In contrast, the DWT obtained a PSNR and RMSE of 17.27 and 1219.30, respectively [24]. The method proposed by Patel requires further validation with more images, from any public multi-modal dataset.

Extensions to the DWT are used to reduce some problems of the DWT, such as discontinuities in the edge and contours presented in two-dimensional signals [40]. One example of these extensions is the NSST, which improves the preservation of multi-dimensional signal features [40]. Based on this, Padma Ganasala and Vinod Kumar developed a framework for multi-modal medical image fusion in 2014 [25]. This framework uses the NSST to obtain the high and low frequency components of an image, to combine them separately, and then reconstruct the fused image using the inverse NSST.

Padma and Vinod tested their method using nine pairs of MRI and CT images, from patients with a severe cardiovascular accident. This method was compared against four different fusion algorithms: (i) image fusion in the Intensity-Hue-Saturation (IHS) space; (ii) Non-Subsampled Contourlet Transform Fusion (NSCT); (iii) image fusion using the NSST in the IHS color space; and (iv) image fusion based on NSCT in the IHS color space. The result showed that the method of Padma and Vinod achieved the best performance in the MI, with a score of 2.99. In comparison, the closest method in performance was the NSCT technique in the IHS color space with an MI of 84.28.

Another example of the NSST for image fusion is the method proposed by Nair et al. in 2021 [29]. Nair et al. used a non-rigid registration with B-splines to align the input images

and used MATLAB to implement the algorithm. For the validation, ten pairs of grayscale and four pairs of color images were used. This method uses objective and subjective performance metrics to validate it against a non-denoising fusion method. Nair et al.'s study demonstrated that denoising the image improves the fusion procedure at the cost of increased execution time.

Another example of MSD, which solves some drawbacks of the DWT, is the proposed method by Na et al. [28]. Na et al. used a Guided Filter (GF), a technique that takes two inputs, an original image and a guided image, to obtain an output image with the information of the input image and some characteristics of the guided image. Na et al. used the DWT to decompose the image and used the GF to improve the weighted maps using the approximation coefficient from the inputs as the guided image. Na et al. validated the procedure with 15 pairs of CT and MRI images against two DWT methods; one using choose-max fusion and the second one using intuitionistic fuzzy inference fusion. The validation was performed using the metrics of the standard deviation, average gradient, and edge strength. The method of Na et al. showed better performance with all the images. However, the method requires further validation because the comparison was performed individually for every image pair, without using any statistical methodology for the selected image sample.

Non-Multi-Scale Decomposition Methods

The non-MSDs are the methods outside the MSD category. Some examples of these are the sub-space and the artificial neural network methods [30,39].

The sub-space methods project a high-dimensional input image into a lower-dimensional space, achieving better efficiency due to the fact that the manipulation of lower-dimensional data requires less memory. A well-known example of sub-space techniques is the Principal Component Analysis (PCA) method [30].

The artificial network methods use a mathematical model to process information, which uses concepts inspired by biological neural networks, where the main advantage is the capability to predict, analyze, and infer the information of a dataset [39].

The Pulse-Coupled Neural Network (PCNN) is one of the most-used NNs, which was developed by Eckhorn, based on the cat visual cortex [41,42].

One example of the PCNN in image fusion is the technique developed by Xu et al., who used PCNN with the quantum-behaved particle swarm optimization [26]. Xu et al. validated the method against five different fusion techniques, using five pairs of MRI and CT images. The compared methods were: (i) Laplacian pyramid; (ii) the PCNN; (iii) the dual-channel PCNN; (iv) the PCNN with differential evolution algorithms; and (v) the dual-channel PCNN with PSO evolutionary learning. In the evaluation, the Xu technique obtained better performance in the MI and Structural Similarity Index (SSIM), scoring 1.72 and 0.77, respectively.

Hybrid Methods

Lastly, some methods use a combination of MSD and non-MSD. These techniques usually have better performance in feature preservation, but have a drawback in the computing times. An example of a hybrid method is the approach of Kong et al., who used the NSST with the PCNN to fuse MRI and CT images [43]. This method was compared against the techniques of the DWT, NSST, PCNN, and NSCT, using the metrics of the RMSE, PSNR, MI, and Structural Similarity Index (SSIM). The results evidenced that the Kong method obtained the best performance, scoring 1.64, 43.84, 0.91, and 0.99 in the RMSE, PSNR, MI, and SSIM, respectively. However, Kong et al.'s method had a drawback in the computation time, with an execution time of 6.279 seconds, which was higher than the method with the best-performing time, the DWT, with an execution time of 0.317 seconds.

Kavitha proposed a hybrid method in 2010, which uses the Integer Wavelet Transform (IWT) and neuro-fuzzy algorithms to combine the wavelet coefficient maps [27]. Kavitha tested his technique against the DWT methods, using two pairs of MRI and CT images.

The results showed that the Kavitha method performed better, with a score of 13.005 and 0.051 in the EN and MI, respectively, which were superior to the DWT score of 5.514 and 0.046, respectively.

### 3.2. Performance Metrics

Neither of the techniques are perfect, and the performance depends on properties such as the imaging modality. For this reason, it is necessary to compare and validate the techniques in specific scenarios. This validation can be subjective or objective: the subjective validation depends on the human visual evaluation, while the objective evaluation uses quantitative performance metrics. Objective metrics are classified into two types: metric-based with the resulting features and metric-based with signal distortion [44]. The first category measures the transference of feature information into the fused image. Examples of these metrics are the MI, SSIM, and the image quality index from Wang and Bovik (Q) [45]. The second category measures the distortion in the fused image, for instance the RMSE, the Standard Deviation (STD), the PSNR, and the Entropy (EN).

### 3.2.1. Entropy

The *EN* estimates the amount of information presented in the images that is calculated with Equation (2), where $P_x$ is the probability of the intensity distribution of the pixel $x$ and $N$ is the number of possible pixel values, 255 for an eight-bit image depth [39,44].

$$EN = -\sum_{x=0}^{N} P_x ln(P_x),\tag{2}$$

### 3.2.2. Mutual Information

This estimates the amount of information transferred from the source image into the fused image. The *MI* is computed with Equation (3) [39].

$$MI(I_i, I_f) = H(I_i) + H(I_f) + H(I_i, I_f),\tag{3}$$

where $I_i$ is the input image, $I_f$ is the fused image, $H(I_i, I_f)$ is the joint entropy between $I_i$ and $I_f$, and $H(I_i)$ and $H(I_f)$ are the marginal entropy of $I_i$ and $I_f$, respectively.

### 3.2.3. Structural Similarity Index

This measures the preservation of the structural information, separating the image into three components: luminance $I$, contrast $C$, and structure $S$. This metric is calculated using Equation (4) [44].

$$SSIM\left(\left(I_i, I_f\right)\right) = \frac{(2\mu_{I_i}\mu_{I_f} + C_1)(2\sigma_{I_i I_f} + C_2)}{(\mu_{I_i}^2 + \mu_{I_f}^2 + C_1)(\sigma_{I_i}^2 + \sigma_{I_f}^2 + C_2)},\tag{4}$$

where $I_i$ is the input image, $I_f$ is the fused image, and $\mu_{I_i}$ and $\mu_{I_f}$ are the mean of $I_i$ and $I_f$, respectively, $\sigma_{I_i}$ and $\sigma_{I_f}$ are the standard deviation of $I_i$ and $I_f$, respectively, and $C_1$ and $C_1$ are constants to avoid instability when $\mu_{I_i}^2 + \mu_{I_f}^2$ or $\sigma_{I_i}^2 + \sigma_{I_f}^2$ is close to zero [46].

### 3.2.4. Universal Image Quality Index

The Universal Image Quality Index (*UIQI*) measures the structural similarities between a source image and the fused image [45]. This metric is a specific form of the *SSIM*, when $C_1 = C_2 = 0$, which is computed with Equation (5) [46].

$$Q = \frac{4\sigma_{I_i I_f}\mu_{I_i}\mu_{I_f}}{(\mu_{I_i}^2 + \mu_{I_f}^2)(\sigma_{I_i}^2 + \sigma_{I_f}^2)},\tag{5}$$

### 3.2.5. Root-Mean-Squared Error

This measures the variance of the arithmetic square root [44]. The *RMSE* is computed with Equation (6) for 2D images.

$$RMSE = \sqrt{\left( \sum_{x=1}^{M} \sum_{y=1}^{N} \left[ I_i(x,y) - I_f(x,y) \right]^2 \right)}, \tag{6}$$

where $I_i(x,y)$ and $I_f(x,y)$ are the pixel value of the input and fused images, respectively, in the position $x$ and $y$, and $M$ and $N$ are the width and the height of the image, respectively.

### 3.2.6. Standard Deviation

The Standard Deviation (*STD*) is the square of the *RMSE* and is computed with Equation (7) [44].

$$STD = RMSE^2, \tag{7}$$

### 3.2.7. Peak-Signal-to-Noise Ratio

The *PSNR* is computed using the *RMSE* with Equation (8) [44].

$$PSNR = 10.Log \left[ \frac{(M \times N)^2}{RMSE} \right], \tag{8}$$

where $M$ and $N$ are the width and the height of the image, respectively.

The suggested method by Maurer et al. is also a powerful tool for the validation of fusion algorithms [47]. This method requires fiducial markers to calculate the Fiducial Registration Error (FRE) and the Target Registration Error (TRE). The FRE measures the distance between corresponding fiducial markers after the registration, while the TRE measures the distance between corresponding points different from the fiducials used for the registration. The measurement of these metrics requires databases with fiducial points, such as the RIRE project [12] or the Non-rigid Image Registration Evaluation Project (NIREP) [48]. Some authors created automatic methods to quantify the registration accuracy, which does not rely on databases. An example is the method proposed by Hauler et al. [22], who used feature detectors to obtain matching points in the images and compute the Euclidean distances between those points. Hauler's method using the Harris detector yielded comparable results to the use of the RIRE dataset, with the drawback of being insensitive to the presence of large misregistration.

### 3.3. Image Fusion Tools

Image fusion is a complex procedure that requires the coding of the registration and merging steps. For this reason, software tools were created to facilitate the implementation of image fusion.

Table 5 shows the main tools in image fusion based on the investigation performed by Keszei et al. [35].

**Table 5.** Image fusion tools.

| Tools | Open-Source | Operating System | Language | Software Platform |
|---|---|---|---|---|
| 3D Slicer | yes | Linux, macOS, Windows | C++ | ITK |
| Advanced Normalization Tools (ANTs) | yes | Linux, macOS, Windows | C++, Python | ITK |
| ART 3dwarper | no | Linux | C++ | N/A |
| Automated Image Registration (AIR) | yes | Linux, macOS, Windows | C | N/A |
| bUnwarpJ | yes | Linux, macOS, Windows | Java | N/A |
| DRAMMS | yes | Linux, macOS, Windows | C++ | N/A |

**Table 5.** *Cont.*

| Tools | Open-Source | Operating System | Language | Software Platform |
|---|---|---|---|---|
| Drop | no | Linux, Windows | C++, Java | N/A |
| Elastix | yes | Linux, macOS, Windows | C++, Python | ITK |
| Flexible Algorithms for Image Registration (FAIR) | no | macOS, Windows | MATLAB | MATLAB |
| FMRIB Software Library (FSL) | no | Linux, macOS, Windows | C++ | N/A |
| FMRIB's Non-Linear Image Registration Tool (FNIRT) | no | Linux, macOS | C++ | FSL |
| Gilles | yes | Linux, macOS, Windows | C++ | N/A |
| Hierarchical Attribute Matching Mechanism for Elastic Registration (HAMMER) | no | Linux, Windows | C | N/A |
| Insight ToolKit (ITK) | yes | Linux, macOS, Windows | C++, Python | ITK |
| Kroon | yes | macOS, Windows | MATLAB | MATLAB |
| Medical Image Processing, Analysis, and Visualization (MIPAV) | yes | Linux, macOS, Windows | Java | N/A |
| Medical Image Registration Toolkit (MIRTK) | no | Linux, macOS, Windows | C | N/A |
| Medical Imaging Toolkit (MITO) | yes | Windows | C++ | N/A |
| NiftyReg | yes | Linux, macOS, Windows | C++ | N/A |
| OsiriX | yes | macOS | C++ | N/A |
| Plastimatch | yes | Linux, macOS, Windows | C++, C | N/A |
| Statistical Parametric Mapping (SPM) | yes | macOS, Windows | MATLAB | MATLAB |
| Symmetric Log-Domain Diffeomorphic Image Registration (SLDIR) | yes | Linux, macOS, Windows | C++ | ITK |

## 4. Discussion

This study provided a literature review about multi-modal image fusion, using articles between January 2010 and April 2021. The main techniques in image fusion and registration were described, and an overview of the main validation metrics and tools used for image fusion was given.

Regarding the first research question, "What are the existing methods of brain image fusion using CT and MRI?", the procedure was divided into two steps: registration and merging. This review found developments in the use of registration to localize electrodes in SEEG. However, the lack of validation data with external objects was found to be a common problem [17,18,20,21]. This problem led to the use of manual registration as a gold-standard for validation, which produced unclear results for the possible subjective errors. The use of datasets such as the RIRE or NIREP for validation can also lead to unclear results due to the lack of external objects in the images. With respect to the image merging techniques, there were methods related to machine learning algorithms. However, there was a lack of developments specific to problems that require external objects in multi-modal imaging.

Concerning the second research question, "What are the tools used to fuse multi-modal brain images?", twenty-two tools were found for the registration and fusion of medical images, and from these, sixteen were open source. The most-used platforms for the development of these tools were ITK and MATLAB, which were used by seven of the found tools (Table 2).

As regards the third research question, "What are the metrics used to validate and compare image fusion methods?", the two main validation methods were: using targets/fiducial points and using specific quantitative metrics. The first one measures the

registration procedure using target points to compute the error between the registered moving image and a gold-standard. The necessity of target points causes difficulty in the error calculation, mainly in problems that lack public gold-standard databases such as SEEG.

From the articles found (Table 2), the image merging methods were validated using metrics based on the resulting features and signal distortion. These metrics measure how the information is transferred to the fused images. In contrast, the registration validations measure the distortion in the resulting image, which commonly requires a gold-standard. To the authors' knowledge, the only gold-standard for multi-modal brain images are the RIRE and NIREP datasets, which cannot be used in specific conditions such as Intracranial Electroencephalography (iEEG). Some authors have used manually registered images [19] or digital photos from the implantation of electrodes in the iEEG to obtain a 3D model of the position of the electrodes and used it as the gold-standard [17,20].

Finally, this review documented the main techniques and tools for the multi-modal fusion of brain images, specifically in the exam of SEEG. There is a lack of merging method for this specific scenario and some problems in the validation methodology for the registration procedure, caused by the lack of a gold-standard dataset with external objects.

## 5. Conclusions

Image fusion is a powerful tool used in the localization of epileptogenic tissue in SEEG [17,18,20,21]. The methods for image fusion in SEEG focus mainly on the registration procedure, with a lack of implementation. This review also revealed that the most common platforms were MATLAB and ITK, which were used in nine of the found articles.

The literature search did not reveal a standard validation method when external objects were present. This lack of standardization complicates the comparison of different fusion techniques, which also causes unclear accuracy results related to electrode localization. Based on this, it is necessary to create a standard validation method for medical images that requires the localization of external objects, including a gold-standard dataset, similar to the RIRE or NIREP.

**Author Contributions:** Conceptualization, J.P., C.M. and M.T.; methodology, J.P., C.M. and M.T.; validation, J.P., C.M. and M.T.; formal analysis, J.P.; investigation, J.P.; resources, J.P.; data curation, J.P.; writing—original draft preparation, J.P.; writing—review and editing, J.P., C.M., M.T. and A.H.; visualization, J.P.; supervision, C.M., M.T., and A.H.; project administration, C.M. and M.T.; funding acquisition, M.T. All authors read and agreed to the published version of the manuscript.

**Funding:** This research received no external funding.

**Institutional Review Board Statement:** Not applicable.

**Informed Consent Statement:** Not applicable.

**Acknowledgments:** Claudia Mazo holds a grant from the Enterprise Ireland EI and from the European Union's Horizon 2020 research and innovation program under the Marie Slodowska-Curie Grant Agreement No. 713654.

**Conflicts of Interest:** The authors declare no conflict of interest.

## Abbreviations

The following abbreviations are used in this manuscript:

| | |
|---|---|
| SEEG | Stereotactic Electroencephalography |
| MRI | Magnetic Resonance Imaging |
| CT | Computer Tomography |
| AED | Anti-Epileptic Drugs |
| MSD | Mean Squared Difference |
| DBS | Deep Brain Stimulation |
| CC | Correlation Coefficient |

| NCC | Normalized Correlation Coefficient |
| MI | Mutual Information |
| NMI | Normalized Mutual Information |
| ITK | Insight ToolKit |
| RIRE | Retrospective Image Registration Evaluation Project |
| TRE | Target Registration Error |
| ERBD | Evolutionary Rigid-Body Docking |
| NGF | Normalized Gradient Field distance |
| MSD | Multi-scale Decomposition Methods |
| LPT | Laplacian Pyramid Transform |
| DWT | Discrete Wavelet Transforms |
| DRT | Discrete Ripple Transform |
| NSST | Non-Subsampled Shearlet Transform |
| RMSE | Root Mean Square Error |
| PSNR | Peak-Signal-to-Noise Ratio |
| HUE | Intensity-Hue-Saturation |
| NSCT | Non-Subsampled Contourlet Transform Fusion |
| Non-MSD | Non-Multi-Scale Decomposition |
| PCA | Principal Component Analysis |
| ANN | Artificial Neural Network |
| PCNN | Pulse-Coupled Neural Network |
| SSIM | Structural Similarity Index |
| IWT | Integer Wavelet Transform |
| EN | Entropy |
| STD | Standard deviation |
| NIREP | Non-rigid Image Registration Evaluation Project |

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
