# Peer review of "MRI and CT Fusion in Stereotactic Electroencephalography: A Literature Review"

_applsci, doi:10.3390/app11125524_

Round 1
Reviewer 1 Report
Overall, the research topic was interesting and the paper contained some useful information.
However, this paper was challenging to understand, especially with too many grammar errors, wrong syntax, and sentence fragments. Therefore, it definitely requires extensive editing. Also, the paper of flow was unorganized. I would definitely advise working on it since this paper cannot be published in its current form. Also, there is a problem with the research method that secondary source (e.g., review) is included for systematic review analysis. For systematic review analysis, a primary source (original research article) should be used for analysis.
Lastly, despite the author's claim, this paper is closer to a literature review than a systematic review, especially when it appears that the paper summarized findings from collected papers rather than generating a synthesis of findings. Also, I could not find an assessment of the validity of the findings of the included studies. Please refer to the following references for example and criteria of systematic: https://www.bmj.com/content/bmj/341/bmj.c3666.full.pdf; https://bmcmedresmethodol.biomedcentral.com/articles/10.1186/s12874-019-0855-0).
My suggestion is to change to 'literature review' for the title of this paper. Otherwise, the paper really requires extensive work before claiming to be a systematic review.
Also, please refer to the attached document for other suggestions.

Reviewer 2 Report
This manuscript performs a systematic review to determine the primary techniques, methodologies, and tools to detect image fusion and relevant performance metrics. However, your review must be systematic based on a state-of-the-art study (significant papers up to 2020) with a clear meta-analysis and synthesis (no narrative review), showing with references the added value. The survey has to be theoretically and critically analyzed. You have to justify your paper selection, your descriptors bibliographically clearly, and the aspects you explore in such a way that your viewpoint is rationalized. Summarize your findings in a table that allows the reader a comprehensive view that guides the corresponding section. A rigorous literature review is not enough; the research gaps it fills have to be significant. Research questions that drive the paper should be built-in, introducing an ongoing and pertinent bibliography (up to 2020). These should be of global interest and not focused on a particular local problem. Identifying a research gap is not enough; the key is showing its significance to the field. Answer your research question in the conclusions; what did we learn compared with current, significant research (up to 2020). The authors should make explicit suggestions about how their study affects the new development in the image fusion field. Is there something new about a particular theory, or is there evidence of theory advancement? However, in addition to the following comments: - Enhance the abstract and conclusion to focus only on the objectives, methodology, and quantitative results. -The text has too long sentences, which makes the meaning unclear. Consider breaking this into multiple sentences as an example in the abstract (L3-L6; L9-L12; L16-L19, etc.). - The language used should adequately inform the reader, and proofreading is mandatory for English grammar and style. The English language, redaction, and punctuation must be improved in general. The manuscript should undergo editing before being submitted to a journal. - Avoid using many references together, such as L22, L169, etc. -Use recent (2021,2020) and unify the references in one style (add/remove pp). - The research paper should be written in the third person's perspective; words such as "we," "our," etc., need to be avoided ( L8, L11, L42, L43). -Use recent and unify the references in one style. - The research paper should be written in the third person's perspective; words such as "we," "our," etc., need to be avoided. -Many grammatical or spelling errors that make the meaning unclear and sentence construction errors, punctuation errors. The following are some examples: L17: worldwide, i.e. around.…… should be …. worldwide, i.e., around L19: tissue, involved in seizure generation, may be.…… should be …. tissue involved in seizure generation may be… L43: the need of the review: .…… should be …. the need for the review:Author Response
Please see the attachment,

Reviewer 3 Report
Undoubtedly, epilepsy is a common neurological disease characterized by spontaneous recurrent seizures. Authors in this article present a systematic review, following the methodology proposed by Kitchenham, to determine the main techniques of multi-modal brain image fusion, the most relevant performance metrics, and the main fusion tools, from these we found a lack in standard validation of the image fusion procedure due to the absence of gold-standard datasets with external objects.
My comments to the article are as follows:
- As part of the Introduction, I propose to provide a broader background in the field of methods of acquisition and analysis of human brain work. For example, reference may be made to: Methods of Acquisition, Archiving and Biomedical Data Analysis of Brain Functioning, Biomedical Engineering And Neuroscience, Book Series: Advances in Intelligent Systems and Computing, Springer, 2018.
- I propose to provide an argument why you chose these and not other databases for exploration?
- References to mathematical formulas should be included in the text.
- Please take into account the plans for the future in the scope of research carried out under Conclusions.
Editing note:
- Tables in the article should be formatted in accordance with the requirements of the MDPI.
Round 2
Reviewer 1 Report
Overall, this manuscript has improved compared to the first draft. However, still, it requires proofreading and editing for grammar checks. Please ensure to change figure 2 since the last square box still states 20 instead of 14 (Note: the second revision only includes 14 original articles). Another suggestion is to include the supplementary document which demonstrates Kitcheman's method actually aligns with the PRISMA guideline. Please check the attached file.

Reviewer 2 Report
The authors did not adopt my main concern related to the motivation and the contribution of this paper. And, unfortunately, they do not present the results of meta-analysis and synthesis (no narrative review), which is the sole of any systematic review. The result section is like a background for the tools used in image fusion (L82-L349), while it should discuss the obtained finding and validate the results supported by graphical and tabular data.
-Research questions that drive the paper should be built-in, introducing an ongoing and pertinent bibliography (up to 2020). Also, these questions should be of global interest and not focused on a particular local problem.
-Answer your research question in the conclusions; what did we learn compared with current, significant research (up to 2020).
-The researcher did not take advantage of improving the quality of research but only made some slight corrections.
-words such as "we," "our," etc., need to be avoided.
-Many grammatical or spelling errors that make the meaning unclear and sentence construction errors, punctuation errors. It requires mandatory proofreading. The following are some examples.
L3-6: Too long sentence and hard to read.
L5: which in turn relies on ……..should be…… which relies on
L13: was not a standard ……..should be…… was no standard
L14: techniques, this caused ……..should be…… techniques; this caused
L15: or this reason it ……..should be…… or this reason, it
Therefore, this work is not suitable for publication in the current form.
Round 3
Reviewer 2 Report
The manuscript is enhanced to a level that could publish in the current form.
Author Response
First, we thank the reviewers for their careful examination of our manuscript. We reviewed the grammar, the spelling typos.